# PeerJ

# An empirical evaluation of four variants of a universal species–area relationship

Daniel J. McGlinn, Xiao Xiao and Ethan P. White

Department of Biology and the Ecology Center, Utah State University, Logan, UT, USA

## ABSTRACT

The Maximum Entropy Theory of Ecology (METE) predicts a universal species–area relationship (SAR) that can be fully characterized using only the total abundance ($N$) and species richness ($S$) at a single spatial scale. This theory has shown promise for characterizing scale dependence in the SAR. However, there are currently four different approaches to applying METE to predict the SAR and it is unclear which approach should be used due to a lack of empirical comparison. Specifically, METE can be applied recursively or non-recursively and can use either a theoretical or observed species-abundance distribution (SAD). We compared the four different combinations of approaches using empirical data from 16 datasets containing over 1000 species and 300,000 individual trees and herbs. In general, METE accurately downscaled the SAR ($R^2 > 0.94$), but the recursive approach consistently under-predicted richness. METE's accuracy did not depend strongly on using the observed or predicted SAD. This suggests that the best approach to scaling diversity using METE is to use a combination of non-recursive scaling and the theoretical abundance distribution, which allows predictions to be made across a broad range of spatial scales with only knowledge of the species richness and total abundance at a single scale.

## INTRODUCTION

The species–area relationship (SAR) is a fundamental ecological pattern that characterizes the change in species richness as a function of spatial scale. The SAR plays a central role in predicting the diversity of unsampled areas (*Palmer, 1990*), reserve design (*Whittaker et al., 2005*), and estimating extinction rates due to habitat loss (*Brooks, da Fonseca & Rodrigues, 2004*). Applications involving the SAR depend strongly on the form of the relationship (*Guilhaumon et al., 2008*) which is known to change with spatial scale (*Palmer & White, 1994*; *McGlinn & Hurlbert, 2012*). Despite the scale-dependence of the SAR, a simple non-scale dependent model (the power-law) is still the most commonly used model for the SAR (*Tjørve, 2003*).

The Maximum Entropy Theory of Ecology (METE) is a unified theory that shows promise for characterizing a variety of macroecological patterns including the species-abundance distribution, a suite of relationships between body-size and abundance, and a number of spatial patterns including the species–area relationship (*Harte et al., 2008*;

Corresponding author
Daniel J. McGlinn,
danmcglinn@gmail.com

*Harte, Smith & Storch, 2009*; *Harte, 2011*). METE adopts the inferential machinery of Maximum Entropy (MaxEnt; *Jaynes, 2003*) to solve for the most likely state of an ecological community (*Haegeman & Loreau, 2008*; *Haegeman & Loreau, 2009*) using only information on the total number of species, the total number of individuals, the total metabolic rate of all the individuals, and the area of the community.

METE predicts that all SARs follow a universal relationship between the exponent of a power-law characterizing the SAR at a particular scale and the ratio of richness and community abundance. The exponent of the SAR is scale dependent, decreasing with increasing spatial scale. Empirical evaluation of the theory suggests that METE is a promising model for the SAR (*Harte et al., 2008*; *Harte, Smith & Storch, 2009*); however, there are currently four different approaches to applying METE to predict the SAR and it is unclear which approach should be used due to a lack of empirical comparison.

There are two distinct versions of METE, recursive (where richness at different scales is obtained by consecutively halving or doubling of area; *Harte, Smith & Storch, 2009*) and non-recursive (where richness at different scales is obtained directly; *Harte et al., 2008*). These two versions predict somewhat different SARs. It is not clear *a priori* which of these versions of METE should be more accurate, and it has been suggested that the best approach should be based on empirical comparisons (*Harte, 2011*). In addition, the METE-SAR is derived using the species-abundance distribution (SAD). The SAD can either be predicted from $N$ and $S$ or the empirical distribution can be used. The most general use of METE for predicting diversity across scales relies on the use of the theoretical abundance distribution, but there have been no comparisons of METE-SAR predictions using theoretical and empirical SADs.

To understand which approach to METE is best for characterizing diversity across scales we conducted a thorough empirical comparison of the four different variants of the METE-SAR prediction: (1) recursive with predicted SAD, (2) recursive with observed SAD, (3) non-recursive with predicted SAD, and (4) non-recursive with observed SAD. Using 16 spatially explicit plant datasets we compared the form and accuracy of the predicted SAR across the four variations of METE at a wide-range of spatial scales and across a diverse set of plant communities with over 1000 species and 300,000 individual trees and herbs.

## METHODS

### Downscaling richness

The METE approach to predicting the SAR is a two-step application of the maximum entropy formalism (MaxEnt): (1) MaxEnt is first used to predict the SAD which represents the probability that a species has abundance $n_0$ in a community of area $A_0$ with $S_0$ species and $N_0$ individuals, $\Phi(n_0|N_0, S_0, A_0)$, and (2) MaxEnt is then used to predict the intra-specific, spatial-abundance distribution which represents the probability that $n$ out of $n_0$ individuals of a species are located in a random quadrat of area $A$ drawn from a total area $A_0$, $\Pi(n|A, n_0, A_0)$. The $\Pi$ distribution is spatially implicit and does not contain information on the spatial correlation between cells. If the observed species abundance

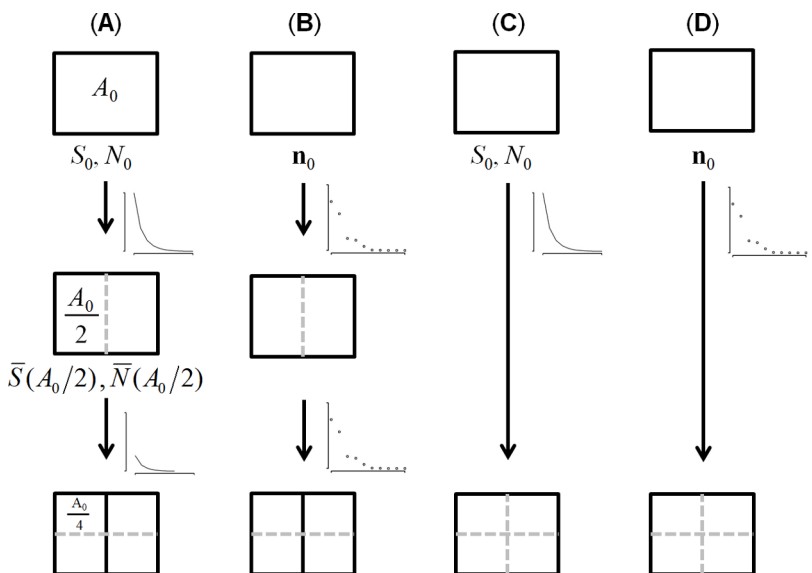

**Figure 1** **An illustration of the process for downscaling species richness from $A_0$ to $A_0/4$ across the four variants of METE.** The recursive approach uses either (A) the theoretical SAD (inset curve) or (B) the observed SAD (inset points) to predict richness at $A_0/2$ and then the process is repeated to generate a prediction at $A_0/4$. In contrast, the non-recursive approach uses either (C) the theoretical SAD or (D) the observed SAD to predict richness at $A_0/4$ directly. $S_0$ is the total number of species, $N_0$ is the total number of individuals, and $\mathbf{n}_0$ is the vector of species abundances at the community scale ($A_0$) abundance.

distribution is used instead of the METE distribution, then only the $\Pi$ distribution is solved for using MaxEnt.

There are no adjustable parameters in METE, and the solutions to $\Pi$ and $\Phi$ only depend on the empirical constraints and possible system configurations (*Haegeman & Loreau, 2008*; *Haegeman & Loreau, 2009*; *Haegeman & Etienne, 2010*). If the observed SAD is not used then constraints on the average number of individuals per species ($N_0/S_0$) and on the upper bound of the number of individuals $N_0$ can be used to yield a truncated log-series abundance distribution (Fig. 1, *Harte et al., 2008*; *Harte, 2011*). To predict $\Pi$, METE places constraints on the average number of individuals per unit area ($n_0A/A_0$) and on the upper bound of the total abundance of a species $n_0$. Although METE requires total metabolic rate to derive its predictions, this variable can be ignored when solving for the METE SAD or SAR (*Harte et al., 2008*; *Harte, Smith & Storch, 2009*; *Harte, 2011*).

There are two ways to downscale (and upscale) the $\Pi$ distribution (Fig. 1). There is a recursive approach (Figs. 1A and 1B) in which the constraints at $A_0$ are used to solve for $\Pi$ at $A_0/2$, which provides new constraints (i.e., predicted $S_{A_0/2}$ and $N_{A_0/2}$) for solving $\Pi$ at $A_0/4$ and so on until richness is computed at every bisection of the total area $A_0/2^i$ where $i$ is a positive integer (*Harte, Smith & Storch, 2009*; *Harte, 2011*, p 159). The recursive approach continually updates its prior information as it downscales richness. Alternatively we can use a non-recursive approach (Figs. 1C and 1D) in which we solve for $\Pi$ at any area based only on the constraints at $A_0$ (*Harte et al., 2008*; *Harte, 2011*, p 243). The recursive approach may be more accurate because it continually upgrades its prior information or

**Peer**J

less accurate due to error propagation thus only empirical comparisons can determine which approach is best used for prediction (*Harte, 2011*, p 160).

*Harte (2011)* provides the derivations for the $\Pi$ and $\Phi$ distributions, so here we will only highlight the most relevant equations for differentiating the four METE variants. The MaxEnt solution to maximizing entropy for $\Pi$ is:

$$\Pi(n|A, n_0, A_0) = \frac{1}{Z_\Pi} e^{-\lambda_\Pi n} \tag{1}$$

where $\lambda_\Pi$ is the Lagrange multiplier and $Z_\Pi$ is the partition function (*Harte, 2011*, Eq. 7.48). The partition function ensures normalization and it is defined as:

$$Z_\Pi = \sum_{n=0}^{n_0} e^{-\lambda_\Pi n} = \frac{1 - e^{-\lambda_\Pi(n_0+1)}}{1 - e^{-\lambda_\Pi}}. \tag{2}$$

The Lagrange multiplier can be solved for by defining $\Pi$ in terms of its constraints which yields:

$$\bar{n} = \frac{n_0 A}{A_0} = \frac{\sum_{n=0}^{n_0} n x^n}{\sum_{n=0}^{n_0} x^n} \left( \frac{x}{1-x} - \frac{(n_0+1) \cdot x^{n_0+1}}{1 - x^{n_0+1}} \right) \tag{3}$$

where $x = e^{-\lambda_\Pi}$ to simplify notation. Although the METE prediction for $\Pi$ can be solved numerically for any area, it is only known analytically for a special case in which the area $A$ is half the total area $A_0$ (*Harte, 2011*, Eq. 7.51):

$$\Pi\left(n \middle| \frac{A_0}{2}, n_0, A_0\right) = \frac{1}{1 + n_0}. \tag{4}$$

Equation (4) shows that METE predicts that all possible arrangements of $n_0$ individuals are equally likely across two equal area quadrats. The flat distribution characterized by Eq. (4) is identical to the prediction offered by the Hypothesis of Equal Allocation Probability (HEAP) model and therefore the recursive application of Eq. (4) to downscale $\Pi$ generates the same set of $\Pi$ distributions as the HEAP model (*Harte et al., 2005*; *Harte, 2007*; *Harte, 2011*):

$$\Pi\left(n \middle| \frac{A_0}{2^i}, n_0, A_0\right) = \sum_{q=n}^{n_0} \frac{\Pi\left(q \middle| \frac{A_0}{2^{i-1}}, n_0, A_0\right)}{(q+1)}, \quad i = 1, 2, 3, \ldots. \tag{5}$$

Thus for a given bisection of the total area (i.e., $A = A_0/2^i$) we can either use the recursive approach (Eq. (5)) or the non-recursive approach (Eq. (1)) to compute the $\Pi$ distribution.

Expected richness is simply the sum of the individual probabilities of species occupancy. Table 1 gives the expressions for expected richness at $A_0/2$ given the four possible combinations of the choice of the downscaling approach and the choice of SAD to use. The equations in Table 1 will also hold for finer spatial scales except for the recursive, theoretical SAD approach which requires downscaling the SAD as well (*Harte, 2011*, Eq. 7.63).

**Table 1** **The four variants of METE formulated for expected species richness at $A_0/2$, given either the recursive or non-recursive method of downscaling and either the theoretical or observed SAD.** These equations also hold for finer spatial scales except for the recursive, METE-SAD approach which requires downscaling the SAD as well (see *Harte, 2011*, Eq. 7.63). $\mathbf{n_0}$ is the vector of empirical abundances and $n_{0,j}$ is the abundance of the $j$th species at the community scale ($A_0$).

|  | Method of downscaling | |
|---|---|---|
| **Species-abundance distribution** | **Recursive downscaling (Eq. (5))** | **Non-recursive downscaling (Eq. (1))** |
| METE-SAD (truncated log-series distribution) | $S_0 \sum\limits_{n_0=1}^{N_0} \left[ \left( 1 - \sum\limits_{q=0}^{n_0} \dfrac{\Pi\left(q\mid \frac{A_0}{2}, n_0, A_0\right)}{(q+1)} \right) \cdot \Phi(n_0\mid N_0, S_0) \right]$ | $S_0 \sum\limits_{n_0=1}^{N_0} \left[ \left( 1 - \dfrac{e^{-\lambda_\Pi \cdot 0}}{Z_\Pi\left(n=0\mid \frac{A_0}{2}, n_0, A_0\right)} \right) \cdot \Phi(n_0\mid N_0, S_0) \right]$ |
| Observed-SAD ($\mathbf{n_0}$) | $\sum\limits_{j=1}^{S_0} \left( 1 - \sum\limits_{q=0}^{n_{0,j}} \dfrac{\Pi\left(q\mid \frac{A_0}{2}, n_{0,j}, A_0\right)}{(q+1)} \right)$ | $\sum\limits_{j=1}^{S_0} \left( 1 - \dfrac{e^{-\lambda_\Pi \cdot 0}}{Z_\Pi\left(n=0\mid \frac{A_0}{2}, n_{0,j}, A_0\right)} \right)$ |

## Empirical comparison

Testing METE's predictions requires spatially explicit, contiguous data from a single trophic level. We carried out an extensive search for data that met these requirements. This search resulted in a database of 16 communities (Table 2; see Table S1 for additional details). All of the datasets are terrestrial, woody plant communities with the exception of the serpentine grassland which is herbaceous. In the woody plant surveys, the minimum diameter at breast height (i.e., 1.4 m from the ground) that a tree must be to be included in the census was 10 mm with the exception of the Cross Timbers and Oosting sites where the minimum diameter was 25 and 20 mm respectively. Where datasets contained time-series information we selected a single census year from each dataset to analyze. *Harte (2011)* suggested that MaxEnt models will perform best when a single process such as the presence of a past disturbance is not dominating the system and rather a multitude of different interacting processes are operating. With this in mind, we attempted to choose the survey years that were the longest amount of time from known stand-scale disturbances (e.g., hurricane events).

For each dataset we constructed fully-nested, spatially-explicit SARs (Type IIA, *Scheiner, 2003*). Recursive METE only makes predictions for bisected areas so we restricted our datasets to areas that were square or rectangular with the dimensional ratio of 2:1. Due to the irregular shape of the Sherman and Cocoli sites we defined two separate 200 × 100 m subplots within each site (Fig. S1). We then calculated the results for each of the two subplots and reported the average.

To assess the accuracy of METE's predictions for the SAD and the four downscaling algorithms of the SAR, we computed the coefficient of determination about the one-to-one line: $R^2 = 1 - \sum_i (obs_i - pred_i)^2 / \sum_i (obs_i - \overline{obs_i})^2$ where $obs_i$ and $pred_i$ are the $i$th log-transformed observed and METE-predicted values (abundance for the SAD, richness for the SAR) respectively. Log transformed richness was used to minimize the influence of the few very large richness values and because relative deviations are of greater interest in evaluating SARs than absolute differences. We used the python package METE (*White et al., 2013*), as well as a suite of project specific R and python scripts for our analysis. All R and Python code used to generate these analyses is archived in the supplemental materials and also available on GitHub (http://github.com/weecology/mete-spatial).

## RESULTS

The four versions of METE all produced reasonable estimates of downscaled richness (Fig. 1). The $R^2$ values ranged from 0.944 for the recursive, observed-SAD model up to 0.997 for the non-recursive, observed-SAD (Fig. 2). Despite the high coefficient of determination, the recursive approach deviated systematically from the empirical data by underpredicting richness (Figs. 2 and 3). This deviation became larger at finer scales (Fig. 2). In contrast, the non-recursive approach showed no systemic deviations. The SAD was well characterized by the METE predictions ($R^2 = 0.95$); however, METE did on average predict slightly more uneven communities (i.e., predicted abundance was too low for rare species and too high for abundant species, Fig. S2). Overall, the

**Table 2 Summary of the habitat type and state variables of the vegetation datasets analyzed.** The state variables are total area ($A_0$), total abundance ($N_0$) and total number of species ($S_0$).

| Site names | Habitat type | Ref. | $A_0$ (ha) | $N_0$ | $S_0$ |
|---|---|---|---|---|---|
| BCI | Tropical forest | a,b,c | 50 | 205096 | 301 |
| Sherman | Tropical forest | d | 2 | 7622.5 | 174.5 |
| Cocoli | Tropical forest | d | 2 | 4326 | 138.5 |
| Luquillo | Tropical forest | e | 12.5 | 32320 | 124 |
| Bryan | Oak-hickory forest | f,g,h | 1.7113 | 3394 | 48 |
| Big Oak | Oak-hickory forest | f,g,h | 2 | 5469 | 40 |
| Oosting | Oak-hickory forest | i | 6.5536 | 8892 | 39 |
| Rocky | Oak-hickory forest | f,g,h | 1.44 | 3383 | 37 |
| Bormann | Oak-hickory forest | f,g,h | 1.96 | 3879 | 30 |
| Wood Bridge | Oak-hickory forest | f,g,h | 0.5041 | 758 | 19 |
| Bald Mtn. | Oak-hickory forest | f,g,h | 0.5 | 669 | 17 |
| Landsend | Old field, pine forest | f,g,h | 0.845 | 2139 | 41 |
| Graveyard | Old field, pine forest | f,g,h | 1 | 2584 | 36 |
| UCSC | Mixed evergreen forest | j | 4.5 | 5885 | 31 |
| Serpentine | Serpentine grassland | k | 0.0064 | 37182 | 24 |
| Cross Timbers | Oak woodland | l | 4 | 7625 | 7 |
| **Ranges** | | | 0.0064–50 | 669–205096 | 7–301 |

**Notes.**

[a] *Condit (1998).*
[b] *Hubbell et al. (1999).*
[c] *Hubbell, Condit & Foster (2005).*
[d] *Condit et al. (2004).*
[e] *Zimmerman et al. (1994).*
[f] *Peet & Christensen (1987).*
[g] *McDonald, Peet & Urban (2002).*
[h] *Xi et al. (2008).*
[i] *Palmer et al. (2007).*
[j] *Gilbert et al. (2010).*
[k] *Green, Harte & Ostling (2003).*
[l] *Arévalo (2013).*

inclusion of the observed SAD did not strongly improve the prediction of the SAR. For the non-recursive approach including the observed SAD improved the overall $R^2$ from 0.984 to 0.997 (Figs. 3C and 3D), but the accuracy of the recursive model actually decreased with the inclusion of the observed SAD ($R^2$ from 0.976 to 0.944, Figs. 3A and 3B).

Results were broadly consistent across datasets, with the exception of the serpentine grassland and Cross Timbers oak woodland. The serpentine community displayed a steeper non-saturating SAR in contrast to the other datasets, and was the only dataset where the recursive downscaling approach was more accurate (Fig. 2O). The oak community displayed a sigmoidal SAR, and in contrast to the other study sites the inclusion of the observed SAD for the oak community resulted in a large improvement in the predicted SAR (Fig. 2P).

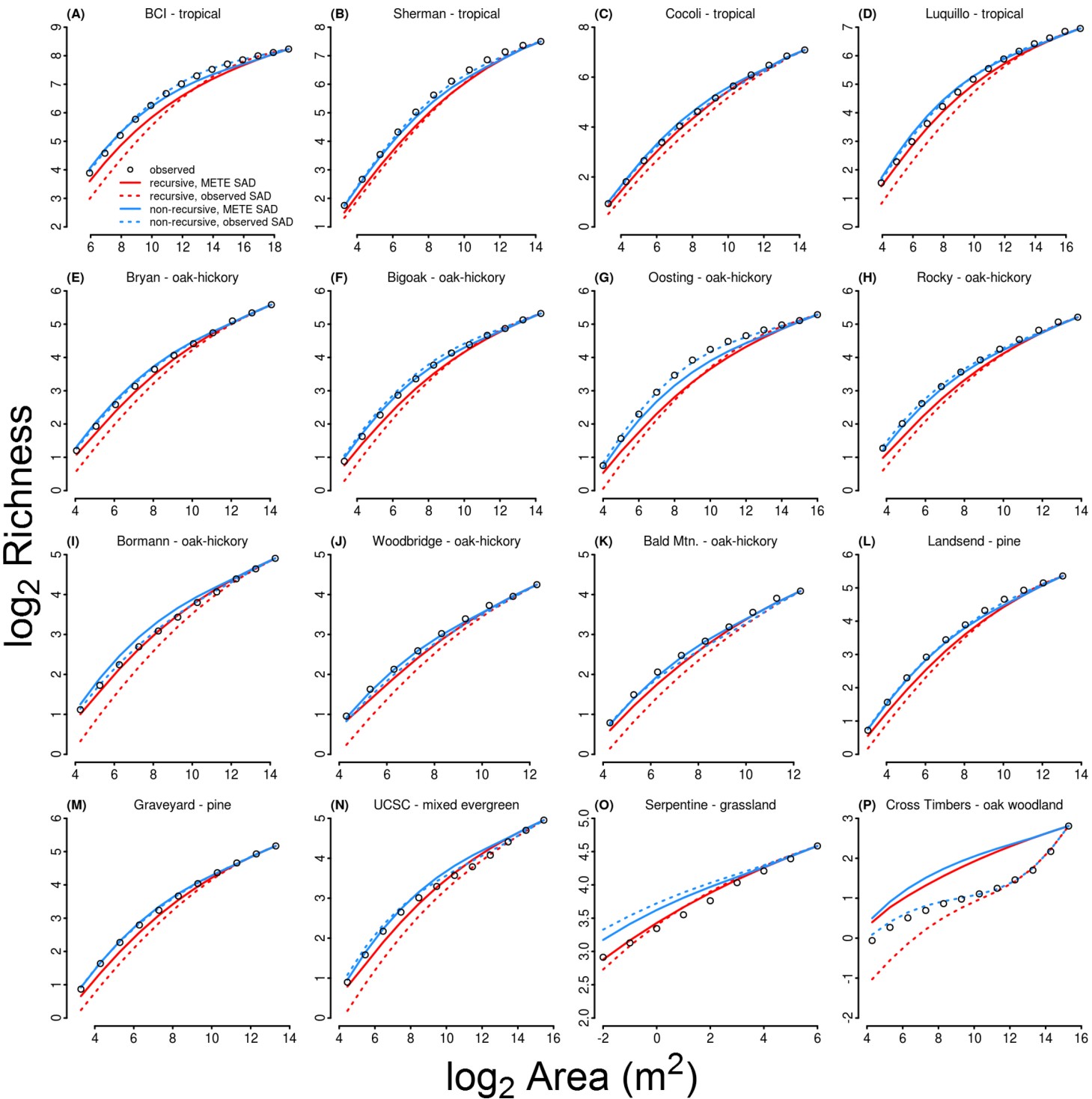

**Figure 2 Empirical species–area relationships and the four versions of the METE model across the 16 sites.** The habitat type of each site is given above each panel. The empirical averages are the open circles, the recursive approach is the red lines, the non-recursive approach is the blue lines, the curves using the observed SAD are dashed and those using the METE-SAD are solid.

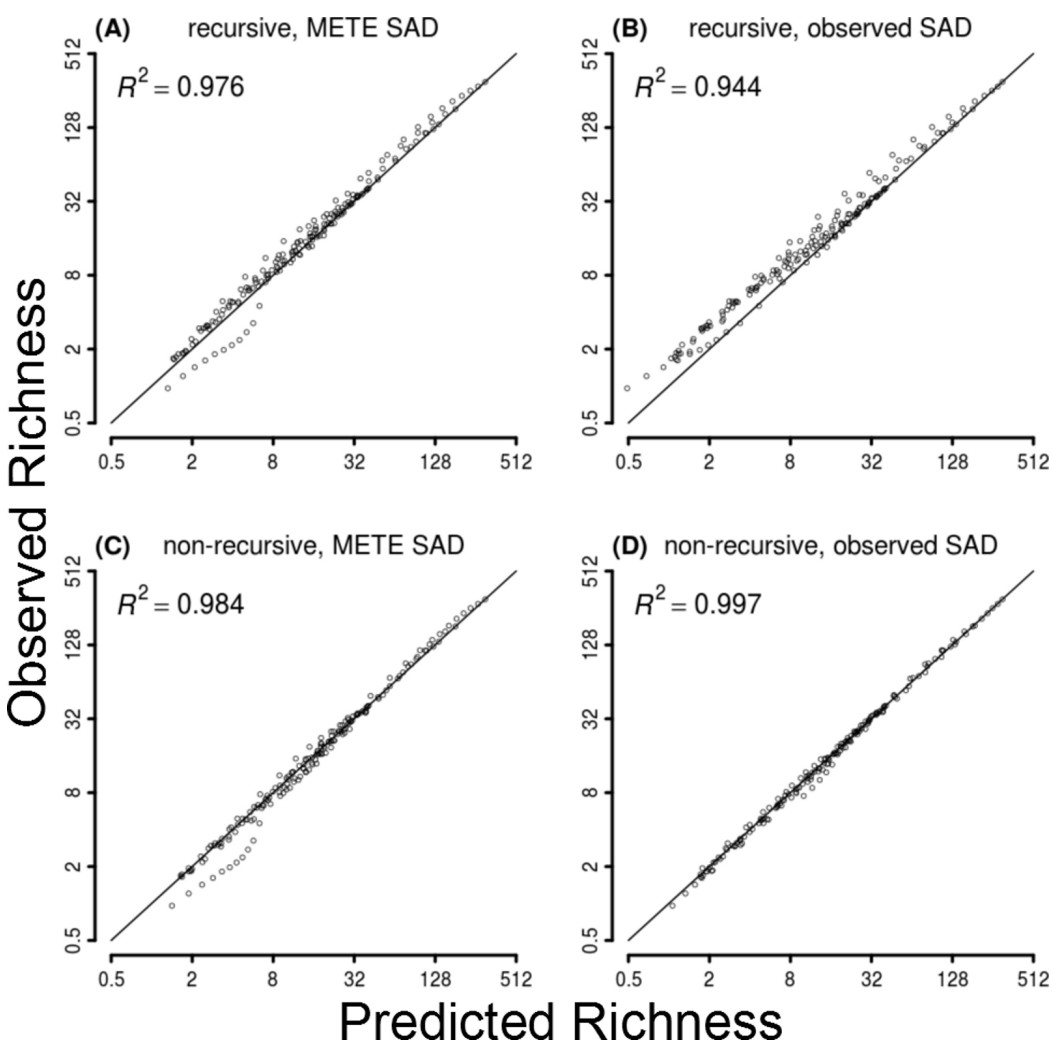

**Figure 3 Observed vs predicted richness across datasets and spatial scales for the four METE SAR models.** The $R^2$-value is computed with respect to the one-to-one line (diagonal).

## DISCUSSION

All four variations of METE performed well at predicting species richness across scales (all $R^2 > 0.94$); however, some versions performed consistently better than others. The non-recursive approach outperformed the recursive version of METE in all but one dataset (the serpentine grassland). The recursive approach also showed small, but consistent, under-predictions for species richness. This means that the recursive approach predicted stronger intra-specific spatial aggregation than observed in the data. This finding is consistent with Harte's (*2011*) comparisons of the species-level spatial abundance distribution in which the recursive approach predicted greater aggregation than the non-recursive approach. Given that the recursive approach provides a poorer fit to empirical data and can only be applied at particular scales (i.e., $A_0/2, A_0/4, \ldots$), we recommend the use of the non-recursive approach for downscaling the SAR. However,

the recursive approach is currently the only means of providing a METE-based prediction for the distance decay relationship via the hypothesis of equal allocation probabilities approach in *Harte (2007)*, and the universal relationship between $S/N$ and the slope of the SAR is currently only known for the recursive approach (*Harte, Smith & Storch, 2009*).

The SAR predictions were generally robust to using the predicted rather than observed SAD. Including the observed SAD increases the amount of information used to constrain the predictions, but it did not substantially increase the overall accuracy of the SAR predictions. This was primarily because the empirical SAD was well characterized by the METE-SAD, consistent with several other studies (*Harte et al., 2008*; *Harte, 2011*; *White, Thibault & Xiao, 2012*). Models in general, and MaxEnt models in particular, typically match empirical data better as increasing numbers of parameters or constraints are included in the analysis (*Haegeman & Loreau, 2008*; *Roxburgh & Mokany, 2010*; *Harte, 2011*). Therefore the naïve expectation for using the observed SAD is that the accuracy of the prediction should increase. However, this was generally only true for the non-recursive approach. This occurred because rarity and intraspecific aggregation interact in subtle ways to determine the shape of the SAR (*He & Legendre, 2002*; *McGlinn & Palmer, 2009*), and simply fixing one of these pieces of information does not guarantee improved predictive power. While using the observed SAD does improve the $R^2$ for the non-recursive form of METE, it only does so by ∼1%. Therefore $N$ and $S$ are generally sufficient to accurately downscale richness using METE across a wide range of habitat types. This is important because it should be possible to model geographic patterns of richness and abundance at a single scale to predict the SAD (*White, Thibault & Xiao, 2012*) and then use those modeled values to predict richness across scales.

Although METE yields accurate predictions for the SAR, its current form has limitations with respect to its extent of applicability and its ability to tie in more broadly with species–time and species–time–area relationships (*Rosenzweig, 1995*; *White et al., 2010*). Specifically, METE predictions are thought to be most relevant for single trophic level datasets that are spatially contiguous and relatively environmentally homogenous (*Harte, 2011*), thus constraining the applicability of METE. At the large spatial scales that are often of interest in conservation planning it is likely that a standard application of METE will fail once species ranges do not occupy all of $A_0$. These are also the scales at which the third phase of the triphasic SAR is expected to occur (*Allen & White, 2003*; *Storch, Keil & Jetz, 2012*), and METE does not predict this accelerating phase. However, *McGill (2010)* suggested that METE's local predictions could be connected with a broad-scale theory to predict a triphasic SAR. Additionally, METE does not currently make predictions through time; however, *Harte (2011)* suggests using Maximum Entropy Production (*Dewar, 2005*). It should be possible to extend METE to predict the species–time–area relationship (*White et al., 2010*) because this pattern, like the SAR, can be modeled in terms of the number of unique individuals sampled per unit area and time (*McGlinn & Palmer, 2009*).

Recently there have been two critiques of the METE spatial predictions. The universality of the relationship between the slope of the recursive METE-SAR and the ratio of $N/S$ was questioned on the basis that the predicted METE-SAR for subsets of a community cannot

be added to yield the community based prediction (*Šizling et al., 2011*; *Šizling, Kunin & Storch, 2013*). However, *Harte et al. (2013)* argue that it is not a flaw of METE or a strong argument against universality.

Additionally, *Haegeman & Etienne (2010)* argued that a multivariate, spatially implicit analog of the univariate Π distribution that is derived using the non-recursive METE approach makes different predictions at different spatial scales (i.e., it is not scale consistent); however, they recognize that a spatially-explicit, scale-consistent version of this distribution may still exist. This critique does not apply to the recursive approach (DJ McGlinn, X Xiao, J Kitzes & EP White, unpublished data), but it may apply in other contexts such as the scaling of the SAD. The lack of scale-consistency in some of METE's predictions suggests that the choice of the anchor scale ($A_0$) may influence the theory's predictions; however, our results, which spanned a range of anchor scales (0.0064 to 50 ha), did not appear to change systematically with scale. Furthermore, *White, Thibault & Xiao (2012)* demonstrated that the METE-SAD accurately characterized empirical SADs across studies with a wide range of anchor scales. Although METE may not provide a universally applicable model of spatial structure in ecological systems and some of its predictions will depend on the anchor scale, our results as well as others suggest that METE can be used as a practical tool for inferring patterns of diversity and abundance from relatively little information.

We examined the down-scaling of richness; however, many conservation applications are interested in up-scaling richness or predicting diversity at a coarse unsampled scale using information at a fine scale. *Harte, Smith & Storch (2009)* demonstrated that recursive-METE accurately up-scaled tropical tree richness. Currently a formal examination of upscaling using the non-recursive approach is lacking. Thus, future investigations should examine the ability of different variants of METE to upscale richness across a range of spatial scales and ecological systems.

METE represents a useful practical tool for accurately predicting species richness across spatial scales. Among METE's four different approaches to predict the SAR, our analysis demonstrates that the non-recursive approach outperforms the recursive approach, and that using the observed rather than predicted SAD does not substantially improve accuracy. Therefore the METE prediction derived using the non-recursive approach and the predicted SAD will likely be the most useful for future applications involving the SAR.

## ACKNOWLEDGEMENTS

Our manuscript was improved by helpful comments from V Bahn, J Harte, J Kitzes, and one anonymous reviewer. We thank RK Peet for providing the oak-hickory and old field pine forest data, J Green for providing the serpentine grassland data, and J Arévalo for providing the oak woodland data.

### Funding

This research was supported by a CAREER grant from the U.S. National Science Foundation to EPW (DEB-0953694). The sources for the data (UCSC Forest Ecology Research Plot, BCI forest dynamics research project, Luquillo Experimental Forest Long-Term Ecological Research Program) had the following funding: The UCSC Forest Ecology Research Plot was made possible by National Science Foundation grants to GS Gilbert (DEB-0515520 and DEB-084259) and by the Pepper-Giberson Chair Fund, the University of California. The BCI forest dynamics research project was made possible by National Science Foundation grants to SP Hubbell: DEB-0640386, DEB-0425651, DEB-0346488, DEB-0129874, DEB-00753102, DEB-9909347, DEB-9615226, DEB-9615226, DEB-9405933, DEB-9221033, DEB-9100058, DEB-8906869, DEB-8605042, DEB-8206992, DEB-7922197, support from the Center for Tropical Forest Science, the Smithsonian Tropical Research Institute, the John D and Catherine T. MacArthur Foundation, the Mellon Foundation, the Small World Institute Fund, and numerous private individuals. The Luquillo Experimental Forest Long-Term Ecological Research Program was supported by grants BSR-8811902, DEB 9411973, DEB 0080538, DEB 0218039, DEB 0620910 and DEB 0963447 from NSF to the Institute for Tropical Ecosystem Studies, University of Puerto Rico, and to the International Institute of Tropical Forestry USDA Forest Service, as part of the Luquillo Long-Term Ecological Research Program. The U.S. Forest Service (Dept. of Agriculture) and the University of Puerto Rico gave additional support. The funders had no role in study design, data collection and analysis, decision to publish, or preparation of the manuscript.

### Grant Disclosures

The following grant information was disclosed by the authors:
NSF: DEB-0953694.

### Competing Interests

EPW is an Academic Editor for PeerJ.

### Author Contributions

- Daniel J. McGlinn conceived and designed the experiments, analyzed the data, contributed reagents/materials/analysis tools, wrote the paper.
- Xiao Xiao and Ethan P. White conceived and designed the experiments, contributed reagents/materials/analysis tools, wrote the paper.

### Data Deposition

The following information was supplied regarding the deposition of related data:
https://github.com/weecology/mete-spatial.

## Supplemental Information

Supplemental information for this article can be found online at http://dx.doi.org/10.7717/peerj.212.

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
