# Peer review of "An empirical evaluation of four variants of a universal species–area relationship"

_PeerJ, doi:10.7717/peerj.212_

## Round 0.1 · original submission · Minor Revisions

· Academic Editor

Minor Revisions

Dear Authors,

Your manuscript has now been reviewed by two experts that are familiar with the METE. Both reviewers are positive about your work and the manuscript and suggest that it can be accepted with minor changes. I am not very familiar with the METE but I have read the manuscript, it is well and clearly written, and has an important message.

I follow the recommendation of the reviewers and ask you to change the manuscript according to the comments of the two reviewers.

In particular I would like to ask you to add a short paragraph about the METE so that the manuscript will be more accessable to readers outside this field, as suggested by reviewer 1. Further, please consider the comments of reviewer 2, and address these concerns in the discussion. Please can you explain the changes you made to the manuscript in a rebuttal, in which you can also provide your arguments in case you disagree with the reviewers.

I appologize for the delay in the returning the manuscript to you. It has been very difficult to find suitable reviewers that were willing to review this manuscript.

Thanks for submitting your work to PeerJ!

·

Basic reporting

A good manuscript on the ability of four slightly differing implementations of a maximum entropy approach to downscale species area relationships. The whole METE approach is great, the antidote to chaos theory so to speak.

The writing and question are clear. I have some minor comments on the manuscript in the detailed comments below. The only general comment I have is that the manuscript could be made a bit more accessible and thus relevant to a broader audience. I’m aware that explanations of METE exist in previous publications cited in the manuscript, but a paragraph on the idea/philosophy of maximum entropy approaches and why and how they work in ecology would really add to the manuscript.

Detailed comments

There are no line numbers in the manuscript

Tables and Figures are messed up. On page 4 you reference Fig 1 but all Figures and Tables are called Tables at the bottom of the document.

Page 6 second paragraph: dbh of 10mm? Really? Shouldn’t that be cm?

Page 7, results – More problems with figuring out the Figures.

Page 7 results, prediction of SAD: maybe additional constraints based on how rare or how abundant species can realistically become would help to improve this.

Experimental design

I don’t see any problems with the science presented, but I have to admit that I did not have the time and ability to go through the math involved. The question and design are clear and solid. Methods are clear and theory reproducible, although that code the authors make freely available would be difficult to use for somebody else.

Validity of the findings

I have no issues with the presentation or soundness of the findings.

Reviewer 2 ·

Basic reporting

The ms is Ok, in my opinion.

Experimental design

The ms is Ok, in my opinion.

Validity of the findings

I like this ms, I have only some suggestions, which, in my opinion, would make conclusions of the ms more fair, accurate and more biologically sound. See the comments for Authors below, please.

Additional comments

The authors present nice and easy to read ms. I have no doubt on the method and the outputs of the computations. I only have some comments to the conclusions. My point of view is that METE is biased by two problems (i) selection of the focal group of species (e.g., one trophic level), (ii) recursive step-by-step computation and (iii) lack of area invariance (i.e., scale inconsistency) for predicted SAD. The authors elegantly eliminated all the problems. They (i) compared results of recursive and non-recursive approach (it is nice to read that non-recursive solution for up-scaling is in progress as not down-scaling, but up-scaling is the main problem, see below) and (ii) avoided the problem with lack of taxon and area invariance by employing the observed SAD.

In my opinion, the authors have shown that the recursive approach biases the solution of METE more than variation between the focal communities (those of the selected by the authors). It is apparent when comparing Figs 5c,d. Via comparing Figs 5a,b one gain information that problem with recursive method and problem with taxon-and-area invariance interact so that both biases act against each other. They have also shown that METE is applicable when one avoids the problems the approach was criticized for by more than one papers. Which is good to know.

On the other hand, I am not convinced that the authors showed that METE is universally applicable and that using observed SAD does not substantially improve the METE results.

Why I am not convinced? Firstly, the authors constrained their datasets to a single trophic level (page 6, 4th line; missing citation at the line), which is exactly what was suggested by Sizling at al. (2011) to avoid the problem with lack of taxon-invariance; however, the Oak woodland data (Fig 4p) suggest that it is not a sufficient condition and thus we need to search for better definition of the community that obey the METE constraints (which is in accord with the discussion Harte et al and Sizling et al. 2013). The case of the oak woodland is very interesting, for I would not expect more cryptic trophic levels there. On the other hand this case remains me the British plant data to which the METE approach also does not apply (competition of up-scaling methods in Leeds; unpublished). Secondly, the fact that observed SAD improved all estimates, including this for Oak woodland, even if data were apriori restricted to a single trophic level, convinced me on the importance of community specific information for METE predictions.

The positive message of the ms, it is that METE is applicable to down-scale biological data. The presented maximum residual is actually less than when some problematic datasets (e.g., the British plant data, BCI) were up-scaled. The case is that METE was also suspected (I am not sure if it was published) that it would not work because live entities (e.g., plants, animals) could mutually interact not to follow the most frequent combination. In this case the ME (maximum entropy) machinery would fail in ecology.

However, in my opinion, this ms is still not a victory (but appreciated step forward) of the METE. The case is that down-scailing is easy task compared with upscaling. (i) Any bias that seems to be negligible when down-scaling is magnified when up-scaling (ii) when up-scaling we enlarge the arena of places with different ecology (perhaps different energetic flows, for METE needs energetically conservative community), and (iii) estimation of large scale SAD based on a small sample is always biased toward ignoring rare species (which is not the case of down-scaling where one observe SAD within a large arena).

(What would the Fig 5 looked like in arithmetic space?. It seems to me that the error converges to zero when approaching eastimates for small areas. The only explanation of mine is now that it is because number of species also approaches zero at small areas. Would not be more accurate to test proportional (in per-cent off estimated species richness) residuals between observed and predicted species richness. This is by the way why down-scaling often seems more accurate than the up-scaling, which is still the big challenge for METE and other theories).

As to the biologically relevant questions: Do the authors thing that the METE represents a picture how the nature really works, or that it is just a practical tool? I would appreciate if the authors may discuss the issue. For example if the METE really captured how the nature work, than I would have a problem with selection biologically relevant spatial scale (spatial anchor) from which we should up- or down- scale species richness (which is one of the points by Haegman and Etiene 2010; see also Sizling2009 PNAS paper, but also Preston 1960 on space and time for SAR paper). If METE is only a practical tool, than the issue of fundamental scale is irrelevant (by the way the SAD prediction by METE is not area invariant thus even in this case the question of fundamental scale may matter; see also Haegman and Etiene 2010).

In conclusion, I like the ms and appreciate it as a nice step forward (whatever the METE will prove applicable to up-scale species richness). I only disagree with some conclusions and I would appreciate (i) more fair discussion with respect of weaknesses of METE and (ii) some author’s opinion on biological relevancy (or if they consider METE as just a practical tool).

Very minor comment: Table 1: The information what the expressions capture is missing from the legend; the summed expressions should be in brackets at the last line of the Tab. (Please checked all the formula again, I am not sure but it seems to me that some typos in indices occurred - unfortunately I am in process of moving and so I do not have literature on top of my finger now).

Good Luck

---

## Round 0.2 · accepted · Accept

· Academic Editor

Accept

dear authors,

I have examined the changes you made to manuscript and the read the rebuttal. I am pleased with the changes and find the manuscript now acceptable for publication.
two minor issues: In none of the files I opened, I did see a title for the paper. This may have been entered somewhere else on the website? Please ensure that the new title is now incorporated.
Second: line 18: should this be relationships?

---

## Author Rebuttal · Round 0.2

Dear Dr. Bezemer,

My coauthors and I would like to thank you and the two reviewers for the helpful feedback on our recent submission to the PeerJ: "An empirical evaluation of four variants of a universal species-area relationship". We have revised the manuscript by changing the title slightly, adding a paragraph to the Introduction that provides a more general and thorough description of METE, and we provided smaller changes throughout the manuscript to address R2's concerns.

Here we will respond to each specific reviewer request:

Response to Volker Bahn:
"The only general comment I have is that the manuscript could be made a bit more accessible and thus relevant to a broader audience. I'm aware that explanations of METE exist in previous publications cited in the manuscript, but a paragraph on the idea/philosophy of maximum entropy approaches and why and how they work in ecology would really add to the manuscript."
--We agreed with Bahn's recommendation and the second paragraph of the Introduction now provides a broader description of METE, how exactly Maximum Entropy is used in the context of the theory, and what this all means in the context of an ecological community.

"There are no line numbers in the manuscript"
--We have added line numbers

"Tables and Figures are messed up. On page 4 you reference Fig 1 but all Figures and Tables are called Tables at the bottom of the document."
--We apologize for the inconvenience and have resolved the table and figure references.

Page 6 second paragraph: dbh of 10mm? Really? Shouldn't that be cm?
--We double checked this and 10 mm is correct.

Page 7 results, prediction of SAD: maybe additional constraints based on how rare or how abundant species can realistically become would help to improve this.
--That is an interesting idea and one that is potentially worth pursuing in a different study.

Response to R2:
"On the other hand, I am not convinced that the authors showed that METE is universally applicable and that using observed SAD does not substantially improve the METE results. Why I am not convinced? Firstly, the authors constrained their datasets to a single trophic level (page 6, 4th line; missing citation at the line), which is exactly what was suggested by Sizling at al. (2011) to avoid the problem with lack of taxon-invariance; however, the Oak woodland data (Fig 4p) suggest that it is not a sufficient condition and thus we need to search for better definition of the community that obey the METE constraints (which is in accord with the discussion Harte et al and Sizling et al. 2013). The case of the oak woodland is very interesting, for I would not expect more cryptic trophic levels there. On the other hand this case remains me the British plant data to which the METE approach also does not apply (competition of up-scaling methods in Leeds; unpublished). Secondly, the fact that observed SAD improved all estimates, including this

for Oak woodland, even if data were apriori restricted to a single trophic level, convinced me on the importance of community specific information for METE predictions."

-- We did not argue that METE is universally applicable across biological systems only that METE predicts all SARs to follow a universal relationship and that Harte et al. (2009) have suggested that METE is universally applicable. In the Discussion, we specifically pointed out that Šizling et al. (2011) criticized the universal claim of Harte et al. (2009). We changed the title of the ms from "An empirical comparison …" to "An empirical evaluation of four variants of a universal species-area relationship" to further emphasize that the universality of the METE-SAR is not fully established.  We agree with R2 that the oak woodland is an example of system specific behavior. We draw attention to this fact in the second paragraph of the Results. The woodland's harsh environment, which is characterized by frequent droughts and a relatively high fire frequency, is likely the cause of the very uneven SAD which diverged strongly from the METE expectation (Supplemental Fig. S2).

We disagree with R2 that "...observed SAD improved all estimates…" because the observed SAD actually decreased the R-squared value of the recursive approach (0.976 to 0.944).  Additionally in the case of the non-recursive approach the observed-SAD only improved the accuracy of the prediction by approximately 1%.  Lastly, we have added text to a paragraph in the Discussion (lines 217-220) that emphasizes that we do not view METE as a universally applicable model of the SAR, specifically we write: "Although METE may not provide a universal model of spatial structure in ecological systems and some of its predictions will depend on the anchor scale, our results as well as others suggest that METE can be used as a practical tool for inferring patterns of diversity and abundance from relatively little information."

"The positive message of the ms, it is that METE is applicable to down-scale biological data. The presented maximum residual is actually less than when some problematic datasets (e.g., the British plant data, BCI) were up-scaled. The case is that METE was also suspected (I am not sure if it was published) that it would not work because live entities (e.g., plants, animals) could mutually interact not to follow the most frequent combination. In this case the ME (maximum entropy) machinery would fail in ecology."

-- The idea that ecological systems will be in their most likely state and thus best characterized by a Maximum Entropy derived prediction when the organisms are not interacting in the system does not appear to be well developed in the ecological literature.  Additionally, METE does not explicitly assume that individuals are non-interacting, although it does treat species neutrally. Therefore, we decided not to address this comment in our revision given that appears to be outside the typical scope and usage of METE.

"(What would the Fig 5 looked like in arithmetic space?. It seems to me that the error converges to zero when approaching eastimates for small areas. The only explanation of mine is now that it is because number of species also approaches zero at small areas. Would not be more accurate to test proportional (in per-cent off estimated species richness) residuals between observed and predicted species richness. This is by the way why down-scaling often seems more accurate than the up-scaling, which is still the big challenge for METE and other theories)."

--In arithmetic space, the observed-predicted plots (which should have been labeled Fig.3 not Fig.5) are very clumped around small values of richness and the overall fit to the data is strongly dominated by the effects of large values of richness. The differences from the one-to-one line in

the log-log observed-predicted plots in Fig. 3 are already displaying proportional deviations. We added text at lines 128-130 that describes why we used log transformed richness.

"As to the biologically relevant questions: Do the authors thing that the METE represents a picture how the nature really works, or that it is just a practical tool? I would appreciate if the authors may discuss the issue. For example if the METE really captured how the nature work, than I would have a problem with selection biologically relevant spatial scale (spatial anchor) from which we should up- or down- scale species richness (which is one of the points by Haegman and Etiene 2010; see also Sizling2009 PNAS paper, but also Preston 1960 on space and time for SAR paper). If METE is only a practical tool, than the issue of fundamental scale is irrelevant (by the way the SAD prediction by METE is not area invariant thus even in this case the question of fundamental scale may matter; see also Haegman and Etiene 2010)."
--We think that METE represents a practical tool for making predictions rather than a model for how ecological systems are actually operating. We added text to the Discussion that makes this clearer, on lines 217-220 we wrote, "Although METE may not provide a universal model of spatial structure in ecological systems and some of its predictions will depend on the anchor scale, our results as well as others suggest that METE can be used as a practical tool for inferring patterns of diversity and abundance from relatively little information."

"In conclusion, I like the ms and appreciate it as a nice step forward (whatever the METE will prove applicable to up-scale species richness). I only disagree with some conclusions and I would appreciate (i) more fair discussion with respect of weaknesses of METE and (ii) some author's opinion on biological relevancy (or if they consider METE as just a practical tool)."
-- We added text in the Discussion that further highlights some of the weaknesses of METE. Specifically, on line 211-215 we drew attention to the fact that the METE-SAD is also likely not scale-consistent and we discussed that the arbitrary choice of the anchor scale could influence the accuracy of METE's predictions. We also made it clearer in the Discussion that we view METE as a practical tool for making predictions.

"Very minor comment: Table 1: The information what the expressions capture is missing from the legend; the summed expressions should be in brackets at the last line of the Tab. (Please checked all the formula again, I am not sure but it seems to me that some typos in indices occurred - unfortunately I am in process of moving and so I do not have literature on top of my finger now)."
-- The caption of Table 1 states that the expressions provide the expectation of richness for $A_0/2$. We added parentheses to make it clear how the summations were to be calculated. Also we corrected a small error in Eq. 1 that also appeared in Table 1 for the non-recursive model, otherwise all of the equations appeared correct.